# Parental knowledge of early childhood caries and its association with bottle-feeding practices: A cross-sectional analysis in the Riyadh Province, Saudi Arabia

Mohammed H. Alshamrani[1]*, Mohammed Ali Kharif[2], Naif Mohammed Alshayiqi[2], Ibrahim Ahmed Alqahtani[2], Belal Diri[2], Hashim Yasser Diri[2], Mohammed Alkithri[2], Mannaa K. Aldowsari[1]

1 Department of Pediatric Dentistry and Orthodontics, College of Dentistry, King Saud University, Riyadh, Saudi Arabia, 2 College of Dentistry, King Saud University, Riyadh, Saudi Arabia

* malshamrani2@ksu.edu.sa

## Abstract

### Background

Early childhood caries (ECC) remains a major public health concern globally and is highly prevalent in Saudi Arabia. Parental knowledge, attitudes, and feeding practices play a pivotal role in ECC prevention; however, data from the Riyadh region remain limited.

### Objective

To assess parental knowledge and awareness of ECC and bottle-feeding attitudes among parents of children aged 1–5 years in the Riyadh region of Saudi Arabia.

### Methods

A cross-sectional survey was conducted among 465 parents using a structured, self-administered questionnaire. The instrument assessed sociodemographic char-acteristics, knowledge of ECC, preventive oral health practices, and feeding behav-iors. Knowledge scores were compared across demographic variables and feeding practices using bivariate analyses. Multivariable linear regression was performed to identify factors associated with knowledge.

### Results

Overall parental knowledge was moderate, with notable gaps in awareness of early dental visits and early signs of ECC. While most parents recognized the importance of brushing primary teeth (82.8%) and restoring decayed primary teeth (65.6%), only 13.1% identified the recommended timing for the first dental visit. Night-time bottle feeding was highly prevalent (73.1%), and prolonged bottle use beyond one year

**Data availability statement:** All relevant data are within the paper and its Supporting information files.

**Funding:** This study was financially supported by the Ongoing Research Funding Program of King Saud University (ORF-2026-2085). No additional external funding was received for this study. The funders had no role in study design, data collection and analysis, decision to publish, or preparation of the manuscript.

**Competing interests:** The authors have declared that no competing interests exist.

was common (72.0%). Higher knowledge scores were significantly associated with brushing after bottle feeding ($p < 0.001$) and early cessation of bottle use ($p = 0.038$). Female gender, higher education, and higher household income were significantly associated with better knowledge ($p < 0.05$), although the model explained a modest proportion of variance ($R^2 = 0.036$).

## Conclusion

Despite moderate overall awareness, substantial knowledge gaps and high-risk feeding practices persist among parents in the Riyadh region. Targeted, early, and culturally appropriate parental education integrated into maternal and pediatric healthcare and supported through digital platforms is essential to reduce the burden of ECC.

---

## Introduction

Early childhood caries (ECC) remains one of the most prevalent chronic diseases affecting infants and preschool children worldwide [1]. The American Academy of Pediatric Dentistry defines ECC as the presence of one or more decayed (cavitated or non-cavitated), missing (due to caries), or filled tooth surfaces in any primary tooth in a child younger than six years [1]. Signs of ECC can occur soon after the eruption of the first primary teeth and often progress rapidly, particularly among children exposed to multiple biological, behavioral, and socioeconomic risk factors [2]. Despite advances in preventive dentistry, ECC continues to be a significant public health burden in both developed and developing countries [3,4].

Untreated caries in early childhood can result in pain, infection, impaired mastication, altered phonetics, and poor nutritional status, thereby negatively affecting children's physical development and overall health-related quality of life (OHRQoL) [5]. Additionally, ECC can have psychological and social consequences for both children and their families, including sleep disturbances, school absenteeism, and increased parental stress [6]. Management of severe ECC is often complex, requiring extensive restorative care or extractions under sedation or general anesthesia, which increases financial costs and carries potential medical risks [7].

Studies have reported that long-term bottle-feeding habit particularly during bedtime, consuming sugary drinks, poor dietary habits, inadequate oral hygiene, and low socioeconomic status have consistently been identified as major contributors to ECC development [8–10]. Reduced salivary flow during sleep further increases the cariogenic potential of nighttime feeding by prolonging carbohydrate exposure and increasing acidogenic bacterial activity, particularly Streptococcus mutans [11]. Although bottle feeding has often been implicated in ECC, evidence suggests that caries etiology is complex and cannot be attributed solely to a single feeding habit [10].

Parental knowledge and attitudes play a pivotal role in shaping children's oral health behaviors. Studies have demonstrated that parents' awareness, beliefs, and practices directly influence their children's oral hygiene routines, dietary patterns, and utilization of preventive dental services [12]. Mothers with higher educational levels

have been shown to possess greater knowledge of oral hygiene practices and the importance of primary teeth compared to those with lower educational attainment [8]. Early dental visits and anticipatory guidance provided to parents are therefore essential in preventing ECC and mitigating its progression [8].

Several studies have reported high ECC prevalence among preschool children, with rates ranging from 70% to over 90% in certain regions [2,4,7]. National and regional data indicate that dietary habits, high sugar consumption, inappropriate feeding practices, and limited parental awareness contribute substantially to this burden [13]. Despite cultural and religious encouragement of breastfeeding, several studies have reported early introduction of bottle feeding and mixed feeding practices among Saudi mothers, often within the first months of infancy [10,14]. Misconceptions regarding milk sufficiency, maternal employment, and lifestyle factors further complicate adherence to recommended feeding guidelines [12].

Although a limited number of studies have explored parental knowledge and awareness of ECC in Saudi Arabia, many were restricted by small sample sizes or single-center designs, underscoring the need for broader, region-specific investigations. Understanding parental knowledge and bottle-feeding attitudes is essential for developing targeted educational interventions and preventive strategies. Therefore, the present study aims to assess the knowledge and awareness regarding ECC and bottle-feeding attitudes among parents residing in the Riyadh region of Saudi Arabia.

## Materials and methods

### Study design and setting

The current cross-sectional questionnaire-based study was carried out following STROBE guidelines. The prior protocol was developed and submitted to the King Saud University ethical committee for approval. The study was carried out for three months from October-December 2025.

The study was conducted at the Dental Hospital, King Saud University Medical City (KSUMC), Riyadh, Saudi Arabia, a major tertiary care center serving a diverse pediatric population. The target population consisted of parents of pediatric patients aged 1–5 years attending KSUMC for dental care. A total sample size of 480 participants was targeted to ensure adequate statistical power and representation of the study population.

### Sample size calculation

The required sample size was calculated using the standard formula for cross-sectional studies:

$$n = \frac{Z^2 \times p(1-p)}{d^2}$$

where Z represents the Z-score corresponding to a 95% confidence level (1.96), p denotes the estimated proportion of parental awareness (assumed to be 50% to maximize variability), and d indicates the margin of error (5%). This calculation yielded a minimum sample size of approximately 384 participants. To compensate for potential non-response or incomplete questionnaires, a larger sample was targeted, and 465 participants completed the survey and were included in the final analysis.

Parents were eligible for inclusion if they were caregivers of healthy children aged between 1 and 5 years and were willing to provide informed consent. Parents of children with congenital, developmental, or systemic medical conditions affecting oral health, as well as those whose children were taking medications known to influence dental-caries risk, were excluded from the study.

The study employed a convenience sampling approach. Eligible participants were approached during their visit to the dental hospital and invited to participate in the study. After obtaining informed consent, participants were asked to complete a self-administered electronic questionnaire. The survey was designed to capture information on bottle-feeding practices, oral-hygiene behaviors, parental knowledge of ECC, and sources of oral-health information.

## Questionnaire development

A structured online-based Arabic questionnaire was developed specifically for this study based on a review of existing literature on ECC risk factors and parental awareness. The questionnaire comprised 26 items distributed across three sections: (1) Demographic Information (8 items); (2) Parental Knowledge of ECC (10 items); and (3) Parental Behavior regarding oral health and bottle-feeding practices (8 items). Responses were recorded using binary or multiple-choice formats. Each correct response to a knowledge item was assigned a score of 1, and each incorrect or "I don't know" response was assigned a score of 0. Higher scores indicate greater parental knowledge of ECC.

The questionnaire was reviewed by subject-matter experts to ensure content relevance and clarity, and minor revisions were made to improve comprehensibility. Content validity was established by a panel of five expert dentists who assessed the relevance and clarity of each item. The Content Validity Index (CVI) for the overall questionnaire was 0.89 (item-level CVI range: 0.81–0.92), and the Content Validity Ratio (CVR) was 0.91. Additionally, the questionnaire was pilot tested on 40 parents to assess clarity and comprehensibility. As no substantial modifications were required, the pilot data were retained and included in the final analysis.

Data were collected electronically using Google Forms, allowing for efficient data entry, storage, and management. Responses were anonymized at the point of collection, and access to the dataset was restricted to the research team to maintain confidentiality.

## Ethical considerations

Ethical approval was obtained from the Institutional Review Board (IRB) of King Saud University Medical City prior to study commencement (IRB Approval No. E-25–9545). Participation was voluntary, and informed consent was obtained from all participants. No personal identifiable information was collected, and all data were handled in accordance with the Declaration of Helsinki.

## Statistical analysis

Data were analyzed using SPSS Software version 24. Descriptive statistics, including frequencies and percentages, were used to summarize demographic variables and parental knowledge levels. Independent samples t-tests were used to compare mean knowledge scores between two-category groups, while chi-square test or Fisher's exact test, where appropriate, was used to assess associations between categorical variables. Multivariable linear regression analysis was conducted to identify factors associated with higher awareness levels. Statistical significance was set at a two-sided p-value < 0.05.

## Results

A total of 465 parents of children aged 1–5 years participated in the study. The majority of respondents were female (59.6%), and 88.8% were Saudi nationals. Participants aged 31–59 years constituted the largest age group (53.3%), followed by those aged 18–30 years (38.7%) and ≥60 years (8.0%). Nearly half of the participants (49.9%) had attained a bachelor's degree, while 38.3% reported a monthly household income of 5,000–10,000 Saudi Riyals (SAR; approximately USD 1,333–2,667) (Fig 1).

Table 1 summarizes parental knowledge regarding ECC and preventive oral-health practices. While a majority recognized the importance of brushing primary teeth (82.8%) and the need to restore decayed primary teeth (65.6%), only about half (51.8%) acknowledged that primary teeth are as important as permanent teeth. Awareness of early preventive dental visits was notably low, with just 13.1% indicating that a child should be taken to the dentist when the first tooth erupts. Knowledge of risk factors was variable; approximately two-thirds identified night-time bottle feeding (63.9%) and early childhood susceptibility to decay (62.8%) as risks.

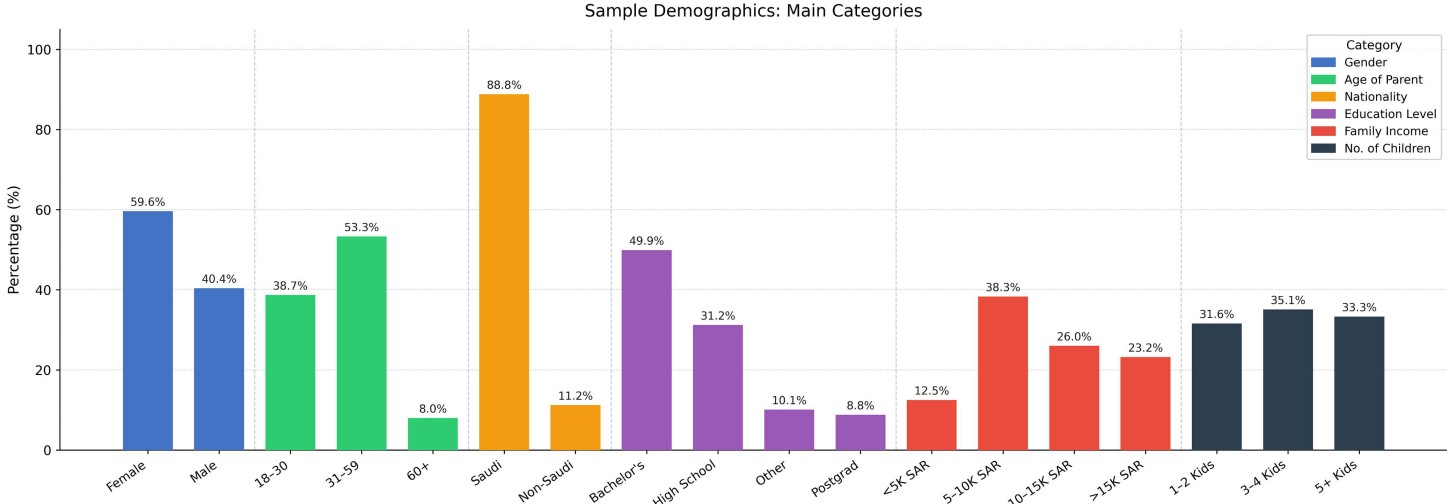

**Fig 1. The main demographic breakdowns for the study sample.**

**Table 1. Parental Knowledge of Early Childhood Caries (ECC) (n = 465).**

| Knowledge Item | Correct Response | n (%) |
|---|---|---|
| Are baby teeth as important as permanent teeth? | Equally important | 241 (51.8) |
| Is it necessary to fill decayed baby teeth? | Yes | 305 (65.6) |
| When do you take your child to the dentist? | First tooth appears | 61 (13.1) |
| Is it important to brush baby's teeth? | Yes | 385 (82.8) |
| Should cleaning start before teeth erupt? | Yes | 269 (57.8) |
| Can decay affect infants below 2 years? | Yes | 292 (62.8) |
| Are white spots on teeth an early sign of decay? | Yes | 266 (57.2) |
| Is decay caused by bacteria transferred via utensils? | Yes | 216 (46.5) |
| Is fluoride in toothpaste important to prevent decay? | Yes | 251 (54.0) |
| Can night bottle feeding cause decay? | Yes | 297 (63.9) |
| **Total knowledge score (0–10)** | | 5.55±2.18 |

n = sample size.

A majority of children had received dental treatment (60.9%). Although most parents avoided sweetened bottles or honey pacifiers (68.8%), nearly one-third continued this practice. Only about half of the parents brushed their child's teeth after bottle feeding (51.6%). Prolonged bottle use was common, with 32.5% continuing for two years or longer, and frequent daily bottle feeding reported by 80.0% of participants. Night-time bottle feeding was highly prevalent (73.1%). While most avoided flavored milk additives (71.8%), delayed stoppage of bottle feeding was observed in 72.0% of children, indicating substantial exposure to cariogenic practices (Table 2).

Parental knowledge of ECC varied significantly by sociodemographic factors (Fig 2, Table 3). Females scored higher than males (5.72 vs. 5.31; p = 0.050). Knowledge was highest among parents aged 31–59 years (5.77±2.03; p = 0.047) and increased with educational background, with postgraduates scoring highest (6.44±2.18; p < 0.001). Household income also correlated positively with knowledge (p = 0.021), whereas nationality showed no significant effect (p = 0.175).

**Table 2.  Parental Behaviors and Feeding Practices (n = 465).**

| Behavior | Category/ Definition | n | % |
|---|---|---|---|
| Child received dental treatment | Yes | 283 | 60.9 |
|  | No | 182 | 39.1 |
| Use of sweetened bottle or honey pacifier | No | 320 | 68.8 |
|  | Yes | 145 | 31.2 |
| Brushing child's teeth after bottle feeding | Yes | 240 | 51.6 |
|  | No | 225 | 48.4 |
| Duration of bottle feeding | ≤1 year | 134 | 28.8 |
|  | 1.5 years | 180 | 38.7 |
|  | ≥2 years | 151 | 32.5 |
| Bottle feeding frequency (per day) | <3 times | 93 | 20.0 |
|  | 3 times | 192 | 41.3 |
|  | >3 times | 180 | 38.7 |
| Night-time bottle feeding practiced | Yes | 340 | 73.1 |
|  | No | 125 | 26.9 |
| Milk with added flavors/additives | No | 334 | 71.8 |
|  | Yes | 131 | 28.2 |
| Age at stopping bottle feeding | ≤1 year | 130 | 28.0 |
|  | >1 year | 335 | 72.0 |

n = sample size.

Brushing after bottle feeding was significantly associated with higher knowledge (6.10 vs. 4.97, t = 5.80, p < 0.001). Early cessation of bottle feeding (≤1 year) also corresponded to higher scores (5.89 vs. 5.42, t = 2.08, p = 0.038). In contrast, avoiding sweetened bottles/pacifiers and nighttime bottle use showed no significant association (p > 0.05) (Table 4).

Multivariable linear regression identified female gender (β = 0.45, 95% CI: 0.05–0.85; p = 0.028), higher education level (β = 0.24, 95% CI: 0.02–0.46; p = 0.036), and higher family income (β = 0.27, 95% CI: 0.07–0.47; p = 0.009) as significantly associated with higher knowledge scores (Table 5). Age group was not significantly associated with knowledge (p = 0.624). The model explained 3.6% of the variance (R² = 0.036), indicating that while sociodemographic factors contribute significantly, other unmeasured determinants may play a larger role in shaping knowledge levels.

## Discussion

The present study provides an updated and region-specific assessment of parental knowledge, awareness, and feeding attitudes related to ECC among parents of children aged 1–5 years in the Riyadh region of Saudi Arabia. Overall, parental knowledge regarding ECC was moderate; however, substantial gaps were identified in critical preventive domains, particularly early dental visits, recognition of early carious lesions, bacterial transmission, and bottle-feeding practices. These findings underscore the need for targeted oral health education programs focusing on early prevention rather than symptom-driven care.

Although most parents recognized the importance of brushing primary teeth and restoring decayed primary teeth, only about half acknowledged that primary teeth are as important as permanent teeth. This misconception has been consistently reported in previous Saudi and regional studies and remains a major barrier to early preventive care [12,15]. The undervaluation of primary teeth may explain the high proportion of parents who sought dental care only after pain or trauma occurred, rather than for routine preventive visits. Notably, awareness regarding the recommended timing of the first dental visit was strikingly low, with only a small fraction indicating that children should visit the dentist upon eruption

 

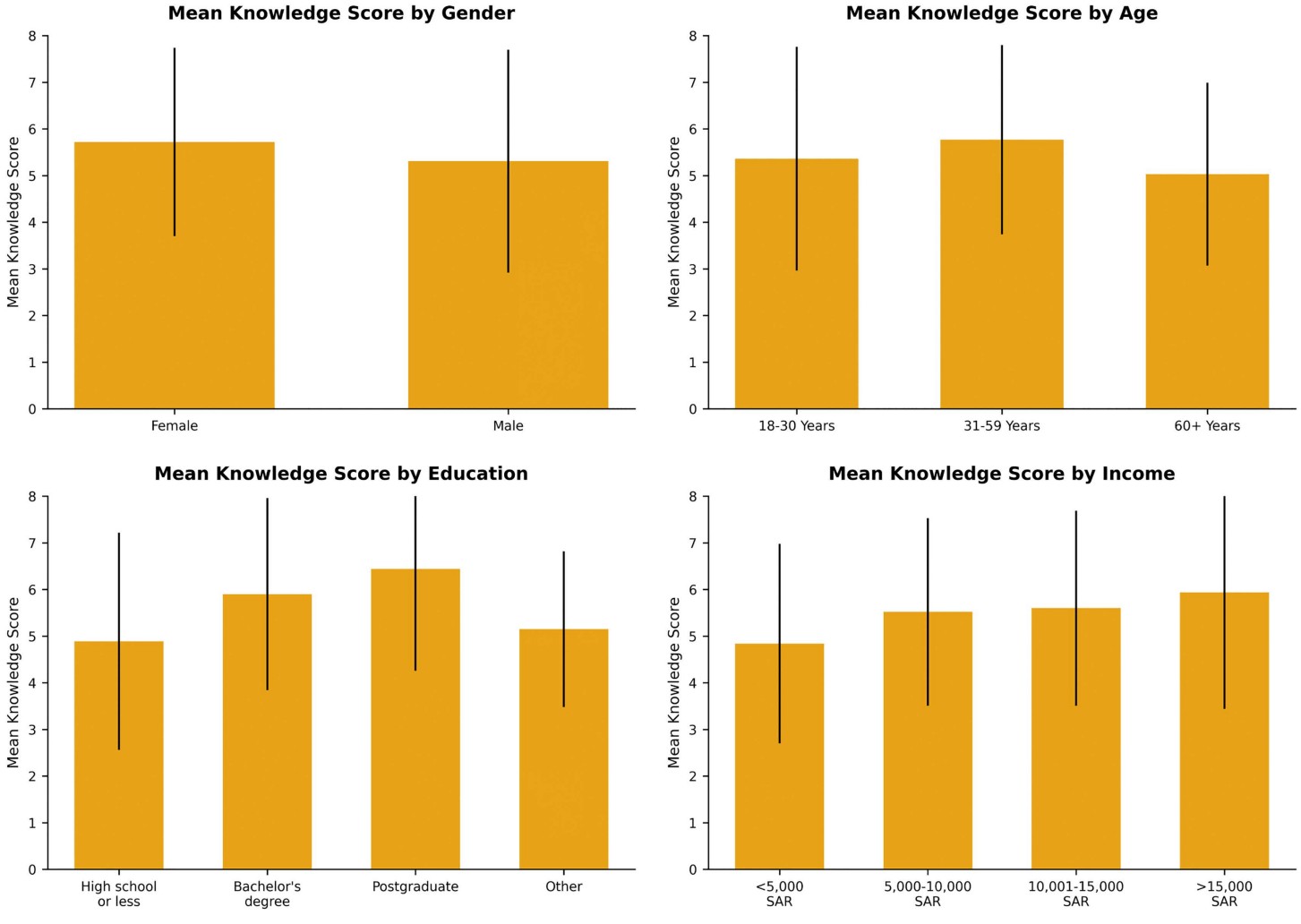

**Fig 2. Association between demographic variables and knowledge score.**

of the first tooth. This finding aligns with previous studies from Saudi Arabia and neighboring Gulf countries, suggesting a persistent gap between professional guidelines and parental awareness [16–18].

Parental knowledge varied significantly by sociodemographic factors. Mothers demonstrated higher knowledge scores than fathers, a finding consistent with prior Saudi studies [4–6], likely reflecting mothers' greater involvement in child care-giving and healthcare utilization. Higher educational attainment has also been associated with greater parental knowledge of ECC and its risk factors [19]. In a meta-analysis, higher odds were reported with caries index and parental education [20]. However, there are no studies reporting parental unemployment rates, income inequality and knowledge regarding ECC [21,22]. These findings reinforce the well-established social gradient in oral health literacy and suggest that educational and economic disparities continue to influence preventive oral health behaviors [15,23]. However, the relatively low explanatory power of the regression model indicates that additional factors such as cultural beliefs, access to oral health information, and health system engagement likely play a substantial role. These results suggest that ECC knowledge in this population may be governed more by cultural and traditional factors than by formal education or income alone.

**Table 3. Bivariate Analysis: Knowledge Score by Demographics.**

| Variable | Groups | Mean±SD | Test | p-value |
|---|---|---|---|---|
| Gender | Female | 5.72±2.02 | t=−1.97 | 0.050 |
| | Male | 5.31±2.39 | | |
| Age | 18–30 years | 5.36±2.40 | F=3.07 | 0.047 |
| | 31–59 years | 5.77±2.03 | | |
| | 60+years | 5.03±1.96 | | |
| Education | High school or less | 4.89±2.33 | F=9.68 | <0.001 |
| | Bachelor's degree | 5.90±2.06 | | |
| | Postgraduate | 6.44±2.18 | | |
| | Other | 5.15±1.67 | | |
| Nationality | Saudi | 5.51±2.21 | t=−1.36 | 0.175 |
| | Non-Saudi | 5.94±1.97 | | |
| Income | <5,000 SAR (<USD 1,333) | 4.84±2.14 | F=3.28 | 0.021 |
| | 5,000–10,000 SAR (USD 1,333–2,667) | 5.52±2.01 | | |
| | 10,001–15,000 SAR (USD 2,667–4,000) | 5.60±2.09 | | |
| | >15,000 SAR (>USD 4,000) | 5.94±2.50 | | |

SD=standard deviation.

**Table 4. Bivariate Analysis: Knowledge Score and Preventive Behaviors.**

| Behavior | Group | Mean Knowledge Score | t-value | p-value |
|---|---|---|---|---|
| Brushing after bottle feeding | Yes | 6.10 | 5.80 | <0.001 |
| | No | 4.97 | | |
| No sweetened bottle/pacifier use | No (never used) | 5.44 | −1.68 | 0.094 |
| | Yes (used) | 5.81 | | |
| No night bottle use | No | 5.46 | −0.59 | 0.55 |
| | Yes | 5.59 | | |
| Stopped bottle at ≤1 year | ≤1 year | 5.89 | 2.08 | 0.038 |
| | >1 year | 5.42 | | |

**Table 5. Multivariable Linear Regression for Factors Associated with Knowledge Score.**

| Variable | Coefficient (β) | 95% CI | p-value |
|---|---|---|---|
| Intercept | 4.56 | 4.03, 5.10 | <0.001 |
| Female gender | 0.45 | 0.05, 0.85 | 0.028 |
| Age group | 0.08 | −0.24, 0.40 | 0.624 |
| Education level | 0.24 | 0.02, 0.46 | 0.036 |
| Family income | 0.27 | 0.07, 0.47 | 0.009 |

Note: Model R-squared=0.036. All factors entered simultaneously. Education was coded as an ordinal variable (high school or less=0, bachelor's degree=1, postgraduate=2, other=3). Income was coded ordinally from lowest to highest category.

Despite acceptable overall knowledge scores, parental attitudes and practices revealed concerning trends. Night-time bottle feeding was highly prevalent, and prolonged bottle use beyond one year was common. Importantly, parents who brushed their child's teeth after bottle feeding and those who discontinued bottle use by one year of age had significantly

higher knowledge scores, suggesting that improved awareness may translate into healthier behaviors. However, the absence of a significant association between knowledge and avoidance of sweetened bottles or night feeding indicates that knowledge alone may be insufficient to modify entrenched feeding practices. Similar discrepancies between knowledge and behavior have been reported in studies from Saudi Arabia, Malaysia, and Kuwait [12,13,24].

These findings raise an important distinction between health knowledge and health literacy. While health knowledge refers to awareness of facts and risk factors, health literacy encompasses the ability to apply that information in real-world contexts. The persistence of high-risk feeding practices despite moderate knowledge levels in this study suggests that health literacy — rather than knowledge alone — may be the missing link between what parents know and what they do. Cultural norms, convenience, and infant-soothing habits may override factual awareness of caries risk. Future interventions should therefore target health literacy and behavioral activation alongside knowledge transfer.

Misconceptions surrounding bottle feeding and breastfeeding were particularly evident. A considerable proportion of parents believed that frequent night-time feeding whether breast or bottle did not contribute to ECC. This misconception has been widely reported in Saudi and international literature [15,16,22]. While breastfeeding remains the optimal nutritional source for infants, prolonged and nocturnal feeding beyond tooth eruption, without appropriate oral hygiene, has been associated with increased ECC risk [24]. These findings highlight the importance of delivering nuanced, evidence-based messages that support breastfeeding while emphasizing oral hygiene after feeding.

Knowledge regarding bacterial transmission through shared utensils or kissing was notably limited, consistent with findings from studies in Brunei, Malaysia, and other regions [18,25]. Given the established role of vertical transmission of cariogenic bacteria from caregivers to children, this represents a critical educational gap. Targeting expectant mothers and caregivers during antenatal and early pediatric visits may be an effective strategy to address this issue.

Parental awareness of fluoride was higher than reported in some earlier Saudi studies, possibly reflecting increased exposure to fluoride-related messaging through commercial advertising and public health campaigns. Nevertheless, misconceptions regarding optimal fluoride use in young children persist. While fluoride remains a cornerstone of caries prevention, inconsistent messaging may contribute to uncertainty among parents, reinforcing the need for clear professional guidance [9]. A substantial proportion of parents recognized white spot lesions as early signs of caries, a finding that contrasts favorably with earlier reports showing minimal awareness [10,11,26,27]. However, recognition alone does not guarantee early intervention, particularly when dental visits remain symptom-driven. Integrating early caries detection messages with guidance on timely dental visits may therefore improve outcomes.

Although dentists remained the primary source of oral health information for many parents, a significant proportion relied on social media. This shift mirrors global trends and presents both challenges and opportunities. In a randomized controlled study, it was reported that mothers found WhatsApp videos useful in understanding oral hygiene maintenance for their children [28]. In a review, a positive trend was seen among dentists to teach mothers the importance of maintaining oral hygiene [21]. Strategic collaboration between oral health professionals, public health authorities, and digital platforms could enhance the reach and effectiveness of ECC prevention campaigns [13,22,24,29].

Studies have reported that ECC is an economic burden on families and society as the treatment requires intensive dental care [14,23,29–32]. However, as primary teeth exfoliate, ECC was not considered an emergency dental disease [33,34]. As studies have been carried out on the prevalence of caries among children above 12 years, there is a scarcity of literature regarding ECC among children below 12 years [19,20,35]. Studies have reported that ECC affects around 49% of preschoolers globally [10,13,15,17,36,37]. Moreover, the distribution of ECC varies according to the geographic areas. A systematic review reported a higher prevalence of ECC in Asian and European countries [21]. Studies reporting the global burden of dental caries have highlighted marked geographic variation in ECC prevalence across regions [33]. Despite increased awareness of preventive strategies, ECC continues to impose a substantial global burden [34]. The findings of the current study indicate that 60.9% of children aged 1–5 years had received dental treatment. While treatment receipt does not directly confirm ECC prevalence, this rate is consistent with the high burden of caries-related dental

disease reported in this age group across the Gulf region, and underscores the need for preventive strategies targeting parents at an early stage.

Several limitations should be acknowledged. The predominance of female respondents may limit generalizability, particularly regarding paternal knowledge and attitudes. Additionally, the use of a self-administered online questionnaire may introduce response and social desirability bias and may underrepresent families with limited internet access. Furthermore, the single-center study design may restrict the generalizability of the results to other regions or healthcare settings. The use of a convenience sampling approach may also limit the representativeness of the study population. As a cross-sectional study, causal inference cannot be established, and regression coefficients should be interpreted as measures of association rather than prediction. Nevertheless, the relatively large sample size compared with prior Saudi studies strengthens the validity of the findings and provides valuable regional insight.

### Clinical implications

Dental health counseling should be integrated into routine pediatric primary care protocols, including the national well-child visit and vaccination schedule. Second, culturally adapted, Arabic-language educational materials targeting parents of young children should be developed and distributed through primary healthcare networks and digital platforms. Third, training programs for primary healthcare providers on guidance for oral health are needed. Finally, national oral health promotion policies should specifically address the early childhood period and integrate structured dental health education into the maternal and child health framework.

### Conclusion

In conclusion, this study highlights persistent knowledge gaps and suboptimal feeding and oral hygiene practices among parents in the Riyadh region, despite moderate overall awareness of ECC. Sociodemographic disparities, misconceptions about bottle feeding, and delayed dental visits remain key challenges. These findings emphasize the need for comprehensive, culturally sensitive oral health promotion strategies targeting parents early in a child's life. Integrating oral health education into maternal, pediatric, and primary healthcare services — particularly through the national well-child visit and vaccination schedule — may be critical to reducing the burden of ECC in Saudi Arabia. Culturally adapted, Arabic-language educational materials should be developed and distributed through primary healthcare networks and digital platforms, and training programs for primary healthcare providers on anticipatory guidance for oral health are warranted. Future research should employ longitudinal designs to determine whether improvements in parental knowledge, achieved through targeted educational interventions, translate into sustained changes in infant feeding and oral hygiene behaviors.

### Supporting information

**S1 File. Data.**
(XLSX)

### Author contributions

**Conceptualization:** Mohammed H. Alshamrani.

**Data curation:** Mohammed Ali Kharif, Naif Mohammed Alshayiqi, Ibrahim Ahmed Alqahtani, Belal Diri, Mohammed Alkithri, Hashim Yasser Diri.

**Formal analysis:** Mohammed H. Alshamrani, Mannaa K. Aldowsari.

**Methodology:** Mohammed H. Alshamrani, Mohammed Ali Kharif.

**Supervision:** Mohammed H. Alshamrani.

**Writing – original draft:** Mohammed H. Alshamrani.

**Writing – review & editing:** Mohammed H. Alshamrani, Mohammed Ali Kharif, Naif Mohammed Alshayiqi, Ibrahim Ahmed Alqahtani, Belal Diri, Mohammed Alkithri, Mannaa K. Aldowsari.

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
