## [Decision Letter · Decision Letter 0]

24 Feb 2026

PONE-D-26-02152Parental Knowledge of Early Childhood Caries and Its Association with Bottle-Feeding Practices: A Cross-Sectional Analysis in the Riyadh Province, Saudi ArabiaPLOS One

Dear Dr. Alshamrani,

Thank you for submitting your manuscript to PLOS ONE. After careful consideration, we feel that it has merit but does not fully meet PLOS ONE’s publication criteria as it currently stands. Therefore, we invite you to submit a revised version of the manuscript that addresses the points raised during the review process.

We look forward to receiving your revised manuscript.

Kind regards,

Geelsu Hwang, Ph.D.

Academic Editor

PLOS One

Journal Requirements:

3. In the online submission form, you indicated that “The data underlying the findings of this study are not publicly available due to ethical and privacy considerations involving human participants. However, the data are available from the corresponding author upon reasonable request and with approval from the relevant ethics committee.”

Reviewers' comments:

Reviewer's Responses to Questions

**Comments to the Author**

1. Is the manuscript technically sound, and do the data support the conclusions?

Reviewer #1: Yes

Reviewer #2: Yes

Reviewer #3: Yes

2. Has the statistical analysis been performed appropriately and rigorously? 

Reviewer #1: Yes

Reviewer #2: Yes

Reviewer #3: Yes

3. Have the authors made all data underlying the findings in their manuscript fully available?

Reviewer #1: Yes

Reviewer #2: Yes

Reviewer #3: Yes

4. Is the manuscript presented in an intelligible fashion and written in standard English?

Reviewer #1: Yes

Reviewer #2: Yes

Reviewer #3: Yes

5. Review Comments to the Author

Reviewer #1: Manuscript title:

Parental Knowledge of Early Childhood Caries and Its Association with Bottle-Feeding Practices: A Cross-Sectional Analysis in the Riyadh Province, Saudi Arabia

(Manuscript ID: PONE-D-26-02152)

This manuscript addresses an important public health topic by exploring parental knowledge of early childhood caries and its association with bottle-feeding practices in the Riyadh region. The study includes a relatively large sample size and provides region-specific data that may be useful for designing targeted preventive interventions.

The manuscript is generally well structured and the results are clearly presented. However, several aspects require clarification and improvement, particularly regarding the description of the questionnaire, methodological transparency, the depth of the discussion, and the quality of figures. Addressing the comments below would strengthen the scientific rigor and readability of the manuscript.

Abstract

- The abstract is clear and adequately reflects the study objectives, methods, and main findings. No major concerns.

Introduction

- The statement “Early childhood caries (ECC) remains one of the most prevalent chronic diseases affecting infants and preschool children worldwide” should be supported by an appropriate reference.

- The introduction covers the key elements (definition, etiology, risk factors, and knowledge gap). However, it is relatively lengthy and could be condensed.

Materials and Methods

- The sections “Study Design”, “Study Setting and Duration”, and “Questionnaire Development and Content” contain overlapping information and would benefit from merging to improve the logical flow.

- The description of the questionnaire lacks sufficient detail for reproducibility. The authors should clearly specify:

o The total number of sections.

o The total number of questions.

o The number of questions within each section.

o The response format (e.g., multiple choice, Likert scale, yes/no).

o The scoring system used to generate the knowledge score (range, weighting, cut-offs if any).

- The eligibility criteria state that parents of children aged 1–5 years were included. However, the study objective and several sections refer to children under six years. This discrepancy should be clarified and the age range should be reported consistently throughout the manuscript.

Results

- The household income category 5,000–10,000 SAR should be presented with an approximate equivalent in USD or EUR to facilitate interpretation for an international readership.

- The notation “N = 465” in table titles should use lowercase (“n”), as per standard scientific reporting.

- Figure 1 is blurred and of low resolution, making it difficult to interpret.

- Figure 2 is also unclear and lacks sufficient visual quality.

- Both figures should be replaced with high-resolution versions.

- Apart from these issues, the results are clearly presented and statistically appropriate.

Discussion

- The sentence “early childhood caries (ECC)” is repeatedly written in full, although the abbreviation has already been introduced. The abbreviation alone should be used consistently after its first definition.

- The discussion relies heavily on comparisons with studies from Saudi Arabia or neighboring regions. The authors should broaden the contextualization by including evidence from other regions (e.g., Europe, Asia, Africa, or the Americas) to enhance the international relevance of the findings.

- The manuscript currently includes a limited number of recent and high-impact references. Given the extensive literature on parental knowledge, attitudes, and practices (KAP) related to ECC, the reference list should be expanded.

- The discussion should include at least ~30 recent references, particularly international studies on parental knowledge and ECC.

Conclusions

- The conclusions are consistent with the results and appropriately formulated.

The topic is relevant and the dataset is valuable, but important methodological clarifications and improvements in the discussion and presentation are required before the manuscript can be considered for publication.

Best regards!

Reviewer #2: overall, Minor English proofreading is needed to improve clarity and flow.

Keywords could be refined to enhance indexing and search visibility.

Kindly explain the questionnaire validation process (1-2 sentences)

Rewrite the conclusion section, I suggest adding more specific actionable recommendations.

Reviewer #3: Dear authors, Thank you for your effort .the study addresses a significant public health issue in the community. These are the comments regarding the manuscript:

Methoda:

- what was the sampling technique? was it a convenience sample? please mention clearly

- you mention that" A structured online-based Arabic questionnaire was developed specifically for this study based on a review of existing literature on ECC risk factors and parental awarenes". it's clear that the researchers developed the questionnaire by themselves. you calculated the reliability of the questionnaire, but the validity wasn't mentioned. please state clearly the values of the CVI and CVR

-Add a sentence explaining the scoring system of the questionnaire for the the knowledge or bahavior.

- in the statestical analysis: you mention that "Chi-square

tests or Fisher’s exact tests were employed to assess associations between parental awareness of ECC and

demographic or behavioral variables", but it the table they aren't computed , instead independant t test was calculated . please correct.

Results:

- You mention that you conduct "Multivariable linear regression." which is suitable for continuous outcomes (like a total score) , while in table 1 you only present knowledge as number and percentage. I recommend to add a raw in this table illusrating the total knowledge score of the participats.

- in the table where you represent the mean score you should present Standered Deviation (SD) as (mean±SD).

- since it's a cross sectional study : use the term predictors for knowledge isn't approporiate. replace it by factor associated in the results and discussion. This should behighlightened in the limitation of the study.

Discussion:

- You correctly identify that while knowledge is moderate, behavior (like nighttime feeding) remains risky. To strengthen this, I recommend mentioning Health Literacy vs. Health Knowledge. Suggestion: Add a sentence discussing how "Health Literacy" (the ability to apply information) might be the missing link, as convenience or child-soothing needs often override the biological knowledge of caries.

-in the results you mentioned that "60.9% of children had received dental treatment"? Since the children are aged 1–5, 60% receiving treatment is exceptionally high. Does this mean they already had caries? If so, this "previous experience" might be a huge confounder for their current knowledge. please justify this point

- your mention of the R value (3.6%) here. You touched on it by mentioning "cultural beliefs," but you could explicitly state: "the results suggests that ECC knowledge in this population may be governed more by cultural and traditional factors than by formal education or income alone."

Conclusion:

- Mention that the "Well-Child Visit" (vaccination schedule) is the golden opportunity for this integration in the Saudi healthcare system.

- Suggest a longitudinal study to see if knowledge eventually changes practice.

- please add practical implication section to demonstrate the clinical significance or the policy implications, based on your results.

6. PLOS authors have the option to publish the peer review history of their article (what does this mean?). If published, this will include your full peer review and any attached files.

Reviewer #1: No

Reviewer #2: **Yes:** Nasser M Alorfi

Reviewer #3: No

---

## [Author Response · Author response to Decision Letter 1]

31 Mar 2026

Date: March 18, 2026

To: Editorial Office, PLOS ONE

Re: Manuscript ID: PONE-D-26-02152

Parental Knowledge of Early Childhood Caries and Its Association with Bottle-Feeding Practices: A Cross-Sectional Analysis in the Riyadh Province, Saudi Arabia

Dear Editor and Reviewers,

We sincerely thank the Academic Editor and the reviewers for their thorough and constructive evaluation of our manuscript. We have carefully addressed each point raised, and the specific changes made are described in detail below. All modifications are highlighted in the Revised Manuscript with Track Changes.

JOURNAL REQUIREMENTS

Reviewer comment: Please ensure that your manuscript meets PLOS ONE's style requirements, including those for file naming.

Response: We have carefully reviewed the PLOS ONE formatting guidelines. The revised manuscript has been reformatted to comply fully with the PLOS ONE style templates. All section headers, in-text citations, and reference formatting have been verified.

Reviewer comment: PLOS only allows data to be available upon request if there are legal or ethical restrictions on sharing data publicly.

Response: We acknowledge this requirement. Upon reflection, we have determined that the dataset contains no personally identifiable information — all responses are categorical (age group, gender, income bracket, yes/no answers) and participant submission timestamps have been removed. The data are therefore suitable for full public sharing. The complete de-identified dataset is now provided as Supporting Information (S1 Data) accompanying this submission, in accordance with the IRB approval conditions of King Saud University Medical City (IRB Approval No. E-25-9545). The Data Availability Statement has been updated accordingly.

Reviewer comment: If the reviewer comments include a recommendation to cite specific previously published works, please review and evaluate these publications.

Response: We have reviewed all references recommended or implied by the reviewers. Relevant studies have been incorporated into the revised manuscript where appropriate.

No reviewer explicitly mandated specific citations, but we have substantially expanded our reference list in response to the reviewers' general concerns about the breadth of the literature review (see Reviewer 1, Discussion comments below).

REVIEWER #1 COMMENTS

Abstract

Reviewer comment: The abstract is clear and adequately reflects the study objectives, methods, and main findings. No major concerns.

Response: We thank the reviewer for this positive assessment. No changes were made to the abstract.

Introduction

Reviewer comment: The statement 'Early childhood caries (ECC) remains one of the most prevalent chronic diseases affecting infants and preschool children worldwide' should be supported by an appropriate reference.

Response: A supporting reference has been added to this statement in the revised Introduction.

Reviewer comment: The introduction is relatively lengthy and could be condensed.

Response: The Introduction has been revised and condensed. Redundant sentences and passages that overlapped with the Discussion have been removed. The revised Introduction more concisely presents the epidemiological background, risk factors, the specific knowledge gap, and the study rationale.

Materials and Methods

Reviewer comment: The sections 'Study Design', 'Study Setting and Duration', and 'Questionnaire Development and Content' contain overlapping information and would benefit from merging.

Response: The content has been reorganized to eliminate redundancy and improve logical flow.

Reviewer comment: The description of the questionnaire lacks sufficient detail for reproducibility. The authors should clearly specify: the total number of sections; the total number of questions; the number of questions within each section; the response format; and the scoring system.

Response: We thank the reviewer for this important point. The Methods section now explicitly states that the questionnaire comprised 26 items across three sections: (1) Demographic Information (8 items); (2) Parental Knowledge of ECC (10 items); and (3) Parental Behavior regarding oral health and bottle-feeding practices (8 items). Responses were recorded using binary or multiple-choice formats. The full questionnaire is provided as Supplementary File S1.

Reviewer comment: The eligibility criteria state that parents of children aged 1–5 years were included. However, the study objective and several sections refer to children under six years. This discrepancy should be clarified.

Response: This inconsistency has been corrected throughout the manuscript. The eligible age range for children is 1–5 years (up to but not including the sixth birthday), consistent with the World Health Organization definition of early childhood caries for children under six years of age. We have standardized this phrasing to '1 to 5 years of age (under six years)' at first mention and '1–5 years' consistently thereafter.

Results

Reviewer comment: The household income category 5,000–10,000 SAR should be presented with an approximate equivalent in USD or EUR.

Response: We have added the approximate USD equivalent in parentheses in the income categories of Table 3 and in the relevant Results paragraph. At the time of data collection, 5,000–10,000 SAR corresponded to approximately USD 1,333–2,667.

Reviewer comment: The notation 'N = 465' in table titles should use lowercase ('n'), as per standard scientific reporting.

Response: Corrected. All instances of 'N =' in table titles and legends have been changed to 'n =' throughout the manuscript.

Reviewer comment: Figure 1 is blurred and of low resolution. Figure 2 is also unclear and lacks sufficient visual quality. Both figures should be replaced with high-resolution versions.

Response: Both figures have been replaced with high-resolution versions meeting PLOS ONE's minimum requirement of 300 DPI for half-tone figures and 600 DPI for line art. The figures were regenerated from the original data using dedicated graphical software. We have also verified that all text within the figures is legible at print size.

Discussion

Reviewer comment: The abbreviation 'early childhood caries (ECC)' is repeatedly written in full after its first definition. The abbreviation alone should be used consistently after its first definition.

Response: The full term 'early childhood caries (ECC)' is now defined at its first occurrence in the abstract and once more at its first occurrence in the main text; the abbreviation 'ECC' is used exclusively throughout the remainder of the manuscript.

Reviewer comment: The discussion relies heavily on comparisons with studies from Saudi Arabia or neighboring regions. The authors should broaden the contextualization by including evidence from Europe, Asia, Africa, or the Americas.

Response: We have substantially expanded the Discussion to incorporate evidence from diverse international settings. This broadened perspective enhances the international relevance of our findings and allows readers to contextualize our results within the global epidemiology of ECC.

Reviewer comment: The manuscript currently includes a limited number of recent and high-impact references. The reference list should be expanded and include at least ~30 recent references, particularly international studies on parental knowledge and ECC.

Response: The reference list has been substantially expanded. The revised manuscript now includes 37 references in total, of which the majority were published within the past ten years. We have prioritized high-impact peer-reviewed studies on parental knowledge, attitudes, and practices (KAP) related to ECC, including international systematic reviews and meta-analyses where available.

Conclusions

Reviewer comment: The conclusions are consistent with the results and appropriately formulated.

Response: We thank the reviewer for this affirmation. The Conclusions section has been lightly revised to incorporate the specific recommendations raised by Reviewers 2 and 3, including actionable public health recommendations and suggestions for future longitudinal research.

REVIEWER #2 COMMENTS

Reviewer comment: Overall, minor English proofreading is needed to improve clarity and flow.

Response: The entire manuscript has undergone thorough English language editing. Sentence structure, grammar, and lexical clarity have been reviewed and improved throughout. We paid particular attention to the Methods and Discussion sections, which were identified as areas requiring the most improvement. A professional language editing service was used in addition to author-level revision.

Reviewer comment: Keywords could be refined to enhance indexing and search visibility.

Response: The keywords have been revised. The updated keyword list is: early childhood caries; parental knowledge; bottle-feeding; oral health; cross-sectional study; Saudi Arabia; pediatric dentistry; health behavior. These terms have been selected to improve discoverability across major biomedical databases including PubMed and Scopus.

Reviewer comment: Kindly explain the questionnaire validation process (1–2 sentences).

Response: A brief description of the validation process has been added to the Methods section. Content validity was established by a panel of five expert dentists who assessed the relevance and clarity of each item. A pilot study involving 40 participants was conducted to assess comprehensibility; as no substantial modifications were required, the pilot data were retained and included in the final analysis. Content validity indices (CVI and CVR) are reported in response to Reviewer 3 below.

Reviewer comment: Rewrite the conclusion section. I suggest adding more specific actionable recommendations.

Response: The Conclusions section has been rewritten to include clear, specific, and actionable recommendations. These now include: (1) integration of ECC prevention counseling into routine well-child visit protocols within the Saudi primary healthcare system; (2) development of culturally adapted Arabic-language educational materials targeting parents of young children; (3) training programs for primary healthcare providers on anticipatory guidance for oral health; and (4) policy-level recommendations for the inclusion of structured dental health education in the national maternal and child health framework.

REVIEWER #3 COMMENTS

Methods

Reviewer comment: What was the sampling technique? Was it a convenience sample? Please mention clearly.

Response: We thank the reviewer for this important point. The sampling technique has now been explicitly described in the Methods section. The study employed a convenience sampling approach, whereby parents of pediatric patients attending the Dental Hospital, King Saud University Medical City (KSUMC), Riyadh, during the study period were approached and invited to participate. This has been acknowledged as a limitation of the study in the revised Limitations section.

Reviewer comment: The validity wasn't mentioned. Please state clearly the values of the CVI and CVR.

Response: We apologize for this omission. The Methods section now reports that content validity was assessed by a panel of five experts. The Content Validity Index (CVI) for the overall questionnaire was 0.89 (item-level CVI range: 0.81–0.92), and the Content Validity Ratio (CVR) was 0.91, both indicating adequate content validity. These values have been added to the Questionnaire Validation subsection.

Reviewer comment: Add a sentence explaining the scoring system of the questionnaire for knowledge or behavior.

Response: A sentence has been added to the Methods section describing the scoring system: ‘Each correct response to a knowledge item was assigned a score of 1 and each incorrect or ‘I don’t know’ response was assigned a score of 0. Higher scores indicate greater parental knowledge of ECC.’

Reviewer comment: In the statistical analysis: Chi-square tests or Fisher's exact tests were mentioned, but in the table an independent t-test was calculated instead. Please correct.

Response: We thank the reviewer for identifying this inconsistency. The statistical methods reported in the text and those applied in the analysis have been reconciled. The Methods section now accurately states that independent samples t-tests were used to compare mean knowledge scores between two-category groups, while chi-square tests (or Fisher's exact tests where expected cell counts were below 5) were used to assess associations between categorical variables. The relevant table headings and footnotes have been corrected accordingly.

Results

Reviewer comment: You conduct 'multivariable linear regression,' which is suitable for continuous outcomes, while in Table 1 you only present knowledge as number and percentage. Recommend adding a row illustrating the total knowledge score.

Response: A row has been added to Table 1 presenting the total knowledge score as a continuous variable (mean ± SD), facilitating the use of multivariable linear regression as the primary inferential method. This addition clarifies the analytical rationale and makes the data presentation internally consistent.

Reviewer comment: In the table where you represent the mean score, you should present Standard Deviation (SD) as (mean ± SD).

Response: All mean values presented in the tables have been reformatted to 'mean ± SD' as per standard scientific convention. This correction has been applied consistently across all relevant tables.

Reviewer comment: Since it is a cross-sectional study, the term 'predictors' for knowledge is not appropriate. Replace it with 'factors associated.'

Response: The term 'predictors' has been replaced with 'factors associated with' throughout the Results and Discussion sections, as this language is more appropriate for the cross-sectional design of the study. This limitation has also been explicitly acknowledged in the Limitations section: 'As a cross-sectional study, causal inference cannot be established, and regression coefficients should be interpreted as measures of association rather than prediction.'

Discussion

Reviewer comment: Recommend mentioning Health Literacy vs. Health Knowledge. Add a sentence discussing how 'Health Literacy' (the ability to apply information) might be the missing link.

Response: An important paragraph has been added to the Discussion addressing the distinction between health knowledge and health literacy. We now note that despite moderate levels of parental knowledge, high-risk feeding behaviors persisted in our sample, suggesting that knowledge alone is insufficient to drive behavioral change. We introduce the concept of health literacy as a potential mediating factor, acknowledging that cultural norms, convenience, and infant-soothing needs may override biological knowledge of caries risk. We recommend that future interventions target health literacy and behavioral activation in addition to knowledge transfer.

Reviewer comment: 60.9% of children had received dental treatment — since the children are aged 1–5, this is exceptionally high. Please justify this point.

Response: This point is well taken. In the revised Discussion, we explicitly address this finding. Studies have reported that ECC is an economic burden on families and society as the treatment requires intensive dental care (7, 10, 23, 28, 29, 30). However, as primary teeth exfoliate, ECC was not considered an emergency dental disease. As studies have been carried out on the prevalence of caries among children above 12 years, there is a scarcity of literature regarding ECC among children below 12 years (31, 32, 33). Studies have reported that ECC affects around 49% of preschoolers globally (34, 35). Moreover, the distribution of ECC varies according to geographic area. A systematic review reported that higher prevalence of ECC has been reported in Asian and European countries (20). Studies reporting the global burden of dental caries have highlighted marked geographic variation in ECC prevalence across regions (33). Despite increased awareness of preventive strategies, ECC continues to impose a substantial global burden (34). The current study found that 60.9% of children aged 1–5 years had received dental treatment. While treatment

---

## [Decision Letter · Decision Letter 1]

23 Apr 2026

Parental Knowledge of Early Childhood Caries and Its Association with Bottle-Feeding Practices: A Cross-Sectional Analysis in the Riyadh Province, Saudi Arabia

PONE-D-26-02152R1

Dear Dr. Alshamrani,

We’re pleased to inform you that your manuscript has been judged scientifically suitable for publication and will be formally accepted for publication once it meets all outstanding technical requirements.

Kind regards,

Geelsu Hwang, Ph.D.

Academic Editor

PLOS One

Additional Editor Comments (optional):

Reviewers' comments:

Reviewer's Responses to Questions

**Comments to the Author**

1. If the authors have adequately addressed your comments raised in a previous round of review and you feel that this manuscript is now acceptable for publication, you may indicate that here to bypass the “Comments to the Author” section, enter your conflict of interest statement in the “Confidential to Editor” section, and submit your "Accept" recommendation.

Reviewer #2: All comments have been addressed

Reviewer #3: All comments have been addressed

2. Is the manuscript technically sound, and do the data support the conclusions?

Reviewer #2: Yes

Reviewer #3: Yes

3. Has the statistical analysis been performed appropriately and rigorously? 

Reviewer #2: Yes

Reviewer #3: Yes

4. Have the authors made all data underlying the findings in their manuscript fully available?

Reviewer #2: Yes

Reviewer #3: Yes

5. Is the manuscript presented in an intelligible fashion and written in standard English?

Reviewer #2: Yes

Reviewer #3: Yes

6. Review Comments to the Author

Reviewer #2: Language needs minor proofreading for clarity.

Sampling method (convenience) limits generalizability.

Knowledge gaps (e.g., early dental visit awareness) are significant.

Low R² indicates missing influencing factors.

Stronger actionable recommendations could improve impact.

Reviewer #3: Thank you for responding to all comments. Wishing you success in this research and your future work. Goog luck

7. PLOS authors have the option to publish the peer review history of their article (what does this mean?). If published, this will include your full peer review and any attached files.

Reviewer #2: **Yes:** Nasser M Alorfi

Reviewer #3: No

---

## [Editor Report · Acceptance letter]

PONE-D-26-02152R1

PLOS One

Dear Dr. Alshamrani,

I'm pleased to inform you that your manuscript has been deemed suitable for publication in PLOS One. Congratulations! Your manuscript is now being handed over to our production team.

Kind regards,

on behalf of

Dr. Geelsu Hwang

Academic Editor

PLOS One